# Exploring context, mechanisms and outcomes in group interpersonal therapy for adolescents with depression in Nepal: a qualitative realist analysis

## Research Article

depression; adolescent; Nepal; interpersonal psychotherapy

**Corresponding author:**
Kelly Rose-Clarke;
Email: kelly.rose-clarke@ucl.ac.uk

Katie H. Atmore[1], Chris Bonell[2], Nagendra P. Luitel[3], Indira Pradhan[3],
Pragya Shrestha[3], Helen Verdeli[4] and Kelly Rose-Clarke[1] (ID)

[1]Institute for Global Health, University College London, London, UK; [2]Department of Public Health, Environments and Society, London School of Hygiene and Tropical Medicine, London, UK; [3]Transcultural Psychosocial Organization Nepal, Kathmandu, Nepal and [4]Department of Clinical and Counseling Psychology, Teachers College, Columbia University, New York City, USA

## Abstract

Interpersonal Psychotherapy (IPT) is an evidence-based treatment for adolescent depression. However, since it does not work for all adolescents in all settings, more research on its heterogeneous effects is needed. Using a realist approach, we aimed to generate hypotheses about mechanisms and contextual contingencies in adolescent group IPT in Nepal. We analysed 26 transcripts from qualitative interviews with IPT participants aged 13–19, facilitators, supervisors and trainers. We analysed data using the Framework Method. The qualitative analytical framework was based on the VICTORE checklist, a realist tool to explore intervention complexity. Sharing, problem-solving, giving and receiving support, managing emotions and negotiating emerged as mechanisms through which adolescents improved their depression. Participants perceived that girls and older adolescents benefitted most from IPT. Girls had less family support than boys and therefore benefitted most from the group support. Older adolescents found it easier than younger ones to share problems and manage emotions. Adolescents exposed to violence and parental alcoholism struggled to overcome problems without family and school support. We formulated hypotheses on group IPT mechanisms and contextual interpersonal and school-level factors. Research is needed to test these hypotheses to better understand for whom IPT works and in what circumstances.

## Impact statement

Psychological interventions including cognitive behaviour therapy and interpersonal psychotherapy (IPT) are evidence-based treatments for adolescent depression, but they do not work for all adolescents in all settings. Surprisingly little is known about how they work, who benefits most (and who benefits least) and in what circumstances they work best. This is important information for policymakers who need to know whether an intervention tested in one place might also be useful and effective somewhere else, and for intervention developers seeking to optimise existing psychological interventions and improve effect sizes. Realist research is an approach to shed light on intervention mechanisms and contextual contingencies that interact with them to influence outcomes. In this study, we describe a novel application of a realist research approach to analyse qualitative data from a feasibility study of group IPT for adolescents with depression in Nepal. We generate hypotheses about mechanisms including sharing, problem-solving, giving and receiving support, managing emotions and negotiating, and individual, interpersonal and school factors that may interact with these mechanisms to bring about reductions in depression. Research is now needed to refine and test our hypotheses in Nepal and explore their relevance in other settings.

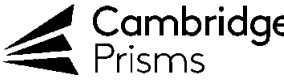

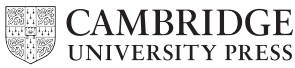

## Introduction

Globally, depression affects 8 % of adolescents aged 10 to 19 (UNICEF, 2021; Shorey et al., 2022). It negatively impacts educational performance and social relationships and increases risk-taking behaviours such as smoking and alcohol use (Thapar et al., 2022). Depression is also a key risk factor for suicide, and physical and mental ill health later in life (Johnson et al., 2018; Thapar et al., 2022).

Psychological interventions including cognitive behaviour therapy (CBT) and Interpersonal Psychotherapy (IPT) are non-pharmacological treatments for depression that focus on the

awareness between mood and depressogenic triggers such as low-reward behaviour and cognitive distortions in CBT, or interpersonal difficulties related to grief, loneliness, life changes and disputes in IPT. Both treatments assist in skills-building for more effective management of these depressogenic triggers. Research mainly from high-income settings indicates psychological interventions work via common therapeutic factors – i.e. general elements common to all types of therapy including attention, hope, and empathy – as well as specific treatment factors related to individual therapies (Wampold, 2015). In Low- and Middle-Income Countries (LMICs) evidence for the effectiveness of psychological interventions is relatively nascent; even less is known about how they work.

According to UK Medical Research Council guidance, psychological interventions are *complex interventions* because they involve multiple components that interact with each other and the context to bring about benefits in mental health (Skivington, 2021). Heterogeneous effects of psychological interventions hint at this complexity: interventions have different effects on adolescents in different settings. For example, evidence from meta-analyses suggests that while overall interventions reduce depression in children and adolescents, effect sizes may be larger in LMICs (effect size 1.01, 95% CI = 0.72–1.29) compared to high-income countries (effect size 0.46) (Eckshtain et al., 2020) (Venturo-Conerly et al., 2023) and factors such as gender, depression severity and history of trauma act to moderate treatment effects (Courtney et al., 2022). There is some research specifically on mediators and moderators of IPT among children and adolescents. For example, two studies report evidence that interpersonal functioning may partially mediate IPT's effects on depression (Reyes-Portillo et al., 2017; Jones et al., 2021). In a trial of group IPT with war-affected adolescents in Uganda, boys and girls with a history of abduction showed significant improvements in depression compared to control, whereas males who had not been abducted did not (Betancourt et al., 2012). In the US a study found that among adolescents with high mother–child conflict at baseline, IPT reduced depression compared to control (regular school counselling), but among adolescents with low baseline conflict IPT had no significant effect (Young et al., 2009).

Realist research could be a useful tool to evaluate psychological interventions because it can shed empirical light on reasons for the heterogeneity of effects and provide insight into potential transferability to other settings (Duncan et al., 2018). Realist research focuses on formulating and testing so-called context-mechanism-outcome configurations (CMOCs) which are hypotheses about how context interacts with intervention mechanisms to generate different outcomes in different populations and settings (Pawson and Tilley, 1997). CMOCs relate to the intervention theory of change and are often informed by stakeholder consultation, existing literature and previous experience in delivering the intervention. Realist methods can be incorporated in traditional randomised controlled trial (RCT) designs which set out to test a priori CMOCs as well as the overall impact of an intervention (Bonell et al., 2012).

In this study, we used a realist approach to explore the complexity of a school-based group IPT intervention for adolescents with depression in Nepal. We used a realist-informed framework to analyse transcripts from qualitative exit interviews with IPT participants and facilitators (Rose-Clarke et al., 2022). Our overall aim was to generate findings that could ultimately inform an intervention theory of change and CMOCs to test in a future realist RCT.

## Methods

### Setting

We analysed data from qualitative interviews conducted in 2020 within the context of a single-arm feasibility study of group IPT involving 62 adolescents aged 13–19. The study was in Sindhupalchowk, a rural district in the foothills of the Nepali Himalayas. According to the 2021 report of the Nepal Statistics Office, the district population is 243,758. The major caste/ethnic groups are Tamang (37%), Chhetri (17%) and Newar (10%) (Nepal Statistics Office, 2021). Sixty-one percent of women and 75% of men aged five and above can read and write. Agriculture is the main source of income. Enrolment rates for primary, lower secondary and secondary school are 95%, 78% and 43% respectively (UN Women, 2016).

### Intervention

IPT aims to empower individuals to understand and manage the interpersonal context of their distress focusing on four "problem areas" (grief, disputes, role transitions and social isolation) (Weissman et al., 2000). IPT was originally developed in the US to treat depressed adults but has been adapted for adolescents and expanded for use in resource-constrained regions (Mufson et al., 1999; Bolton et al., 2007; World Health Organization and Columbia University, 2016; Thurman et al., 2017). In Nepal, we delivered IPT to single-sex groups of six to eight adolescents in secondary schools (Rose-Clarke et al., 2020). The intervention, described elsewhere (Rose-Clarke et al., 2022), involved 12 weekly group sessions plus two pre-group sessions. Pre-group sessions aimed to establish rapport, gather information about important relationships and the depressive symptoms, identify the relevant problem area(s), and mobilise support from the adolescent's family. The facilitator met with the adolescent alone in school for the first pre-group session and with the adolescent and their parent at home for the second. The first group session focused on reviewing and sharing problems and instilling hope for recovery. In sessions two to 11, adolescents learned and implemented interpersonal skills to address their problems and supported other group members to resolve theirs. In the last session, adolescents celebrated their progress and made plans to address any future problems. The facilitators were three staff nurses and three individuals aged 20–25 from the local community, who worked in pairs. They received 10 day's training on basic psychosocial and group management skills and 10 days on IPT, followed by practice with small groups of adolescents. Facilitators were supervised in person and remotely by two Nepali supervisors who were in turn supervised by master trainers in the US and Israel.

### Data collection

We analysed data from semi-structured interviews (SSIs) with 16 adolescents who participated in IPT and all six IPT facilitators. SSIs with facilitators were conducted after the termination of group IPT sessions and with adolescents eight to 10 weeks after IPT. Interviews aimed to explore experiences of participating in and facilitating IPT. Topic guides included questions on acceptability (E.g. *How did you feel about being asked to join IPT?*), utility (*Who is IPT helpful for? What kinds of problems is IPT helpful for? What were the most helpful parts of IPT?*), safety (*Did IPT have any bad effects on you?*) and implementation (*What did you think about the number of IPT sessions?*). We used the Nepali term *man ko samasya* (heart-mind problem) to ask about depressive symptoms because this is a locally acceptable way to communicate feelings of sadness

or depression without incurring social stigma (Kohrt and Harper, 2008). Moreover, this term was used by facilitators and in intervention materials (Rose-Clarke et al., 2020).

SSIs were conducted in Nepali by two Nepali researchers, audio-recorded, transcribed verbatim and translated into English. SSIs with adolescents were in person in their homes. SSIs with facilitators were online because of COVID lockdown restrictions at the time these interviews were conducted.

From the 62 feasibility study participants, adolescents were purposively sampled for SSIs to represent age, gender and caste/ethnicity and low (one to four sessions), medium (five to eight sessions) and high (nine to 12 sessions) attendance of the groups (Table 1).

In 2023, to collect more specific data on intervention mechanisms and complexity we conducted interviews with the two IPT supervisors and two master trainers. We used a more direct approach to explore CMOCs, and topic guides included questions on outcomes, mechanisms and contextual factors (Bonell et al., 2022). Interviews were conducted online in English by KA and transcribed using an automated transcription function.

## Analysis

KA and KR-C analysed data using the Framework Method (Gale et al., 2013). We developed an analytical framework – a list of codes and their definitions – informed by the VICTORE Complexity Checklist (Pawson, 2013). The checklist is used in realist evaluation to explore intervention complexity by breaking it down into components which can be individually considered to promote understanding of the intervention mechanisms (Cooper et al., 2020). VICTORE is an acronym for the following components: *volitions* (participants' reasoning and decisions); *implementation* (processes to operationalise the intervention); *context* (characteristics that impact the programme); *time* (how history and timing affect programme outcomes); *outcomes* (the consequences of a programme); *rivalry* (the impact of other programmes); and *emergence* (unexpected outcomes or events). Our analytical framework included deductive codes for each of the VICTORE components. Table 2 presents Pawson's original definitions of the components and our modifications to make them more relevant to our study. KA coded the transcripts line by line, coding the text under headings from the framework. Coded sections were summarised and charted into a matrix. KA and KR-C reviewed the matrix, using it to compare and contrast data and identify sub-

**Table 1.** Adolescent and facilitator characteristics

|  | Adolescents n = 16 (%) | Facilitators n = 6 (%) |
|---|---|---|
| Median age in years (IQR, range) | 15.5 (15–16, 14–18) | 24.5 (24–26, 20–27) |
| Gender n (%) | Female 8 (50) Male 8 (50) | Female 3 (50) Male 3 (50) |
| Ethnicity/caste n (%) | -Brahman 3 (19) -Chhetri 4 (25) -Dalit 3 (19) -Janjati 6 (38) | -Brahman 1 (17) -Chhetri 1 (17) -Dalit 4 (66) |
| School class n (%) | Class 8 to 4 (25) Class 9 to 10 (63) Class 11 to 2 (13) | |
| Session attendance n (%) (1–4 = low, 5–8 = medium, 9–12 = high) | Low 3 (19) Medium 5 (31) High 8 (50) | |

**Table 2.** Definitions and interpretations of VICTORE checklist components

| Code | Original definition (Pawson, 2013) | Interpretation for the Analytical Framework |
|---|---|---|
| Volitions | The choices subjects make to achieve the ambitions of the programme, including the pathways of persuasion and sequence of choices a subject makes to move from outsider to insider status. | What drove adolescents to participate in IPT and the choices they made. |
| Implementation | The implementation chains of the programme include flows of resources, chains of responsibilities, and reception and transmission points for subjects. | The processes through which IPT was operationalised such that it led to changes in individuals' thoughts, feelings and actions, and/or between individuals and their environment in terms of agency and structure. |
| Contexts | Pre-existing contexts in which the programme is embedded, considering for whom and in what circumstances the programme might work. | The environment in which IPT was implemented, considering family, school and community contexts. |
| Time | The history of the family of programmes of which the intervention under study is a member, considering how what happened before will shape what happens next. This includes previous experiences of programme subjects and communities with similar programmes | Existing adolescent and community concepts of depression and experiences with mental health services, and how these may have shaped their engagement and participation in IPT. |
| Outcomes | The monitoring systems applied include measures likely to be contested, how stakeholders might differ in their interpretations, and whether behaviour might change as a result of being monitored rather than as a result of the intended action of the programme. | Changes perceived and attributed to participation in IPT by stakeholders, including positive and negative changes at the micro to macro levels. |
| Rivalry | The pre-existing policy landscape in which the programme is embedded. Other, contiguous programmes and policies may share or oppose the ambitions of the intervention under study and can override the actions of stakeholders and subjects under study. | Conflicting priorities prevent adolescents from engaging in IPT. |
| Emergence | The potential emergent effects, long-term adaptations, societal changes and unintended consequences associated with the programme, considering whether the spread and duplication of the programme might blunt its effective-ness. | Emerging effects and issues arising during IPT prompted further adaptation of the intervention. |

themes. We discussed emerging trends and KR-C double-coded a portion of the transcripts to resolve ambiguities. Findings were shared and revised with supervisors and trainers who are included as co-authors (IP, PS and HV) to recognise their contribution to the process.

### Ethics

Nepal Health Research Council (637/2018) and King's College London Research Ethics Committee (KCL REC, RESCM-18/19–8,427) approved the feasibility study (including qualitative interviews with adolescents and facilitators). We obtained further ethical clearance for interviews of IPT supervisors and master trainers in 2022 from KCL REC (MRA-22/23–34,693). We obtained informed consent for all participants and parental consent for adolescents under 18.

### Findings

Using the VICTORE checklist, we explored the complexity of group IPT for adolescents with depression in rural Nepal. Our findings are presented under each component of the VICTORE checklist.

### Volitions: Why adolescents chose to participate in IPT and continued to attend IPT sessions

> I would only participate if it's fun. Also, I wouldn't join if I didn't have heart-mind problems. Male, 15.

Adolescents chose to participate in IPT because they perceived they had a heart-mind problem and that IPT would be beneficial ("I felt the training would be good for me"). Moreover, they were influenced by family and friends ("We were told by older people that it is helpful for our heart-mind"). Adolescents believed that facilitators wanted to help them and liked that facilitators spoke to them with respect ("I can tell he is a good person"). Participants perceived IPT to be an opportunity to "learn, know, listen to and understand good things".

Adolescents identified several reasons for continuing to attend IPT. They described the strategies they learned in sessions as useful. Adolescents said they attended because they enjoyed the sessions and had fun. Supervisors and facilitators thought that adolescents participated in IPT because it helped them build connections with peers, reduce loneliness and isolation, and be more successful in school. Facilitators thought adolescents valued the experience of feeling supported and learning to support others in a safe environment but that they would stop attending once their problems were resolved.

Three adolescents with low attendance stopped attending sessions because (i) they wanted to practice for a football competition; (ii) they felt sessions became repetitive; and (iii) they were worried about missing classes and falling behind with schoolwork. A 16-year-old girl with high attendance said she felt "irritated" that she had to continue to attend sessions after her problem was resolved.

### Implementation: Processes through which IPT was operationalised and impacted outcomes

*Sharing, problem-solving, giving and receiving support*, and *managing emotions and negotiating* emerged as key processes through which IPT helped to improve adolescents' heart-mind problems.

### Sharing

All adolescents recognised the value of sharing their thoughts and feelings, and that they would not usually share such things outside IPT. Adolescents described how sharing problems and listening to others helped them to normalise their experiences and brought relief. One adolescent said that compared to individual sessions, group sessions were better because of the opportunity to share with others. It took adolescents varying amounts of time to feel comfortable sharing. A facilitator described how it took adolescents whose problem area was grief or loneliness longer to share than those whose problem was a role transition or dispute.

A facilitator described how adolescents felt uncomfortable discussing suicide. One adolescent described how they would prefer one-on-one sessions because they were uncomfortable sharing in a group.

For sharing to occur, adolescents said they needed confidentiality and a "peaceful" space to feel safe. Adolescents were initially worried that what they said in groups would be "leaked" and that facilitators would disclose information to their parents. They worried that a confidentiality breach could harm their family's reputation (*ijjat*). Adolescents and facilitators described how group rules about not sharing discussions with people outside the sessions were endorsed by all participating adolescents and helped to address some of the confidentiality concerns. Group rules helped adolescents to feel safe, especially rules about active listening and "speaking one-by-one".

Facilitators tried to ensure all group members had an opportunity to speak and created a safe environment for sharing.

> *They understood us, they weren't like others who get lost and ignore us while speaking or divert the conversation in between. They weren't like our teachers but were like friends. They were fun and they made it easy to express everything. They listened to us and provided suggestions. That's why I liked them a lot. Female, 16.*

### Problem-solving

Generating and implementing solutions emerged as key processes that enabled adolescents to resolve their heart-mind problems. In sessions, adolescents listened to each other's problems and explored ways to solve them. They perceived the need to decide for themselves which solution would work best, test it out in the group through role play and implement it later at home, school, or in the community. Problem-solving was described as an iterative process: adolescents would implement a solution and if it was not successful the group would discuss alternatives. For example, an adolescent was beaten by her father. The adolescent tried to speak to her father but was verbally abused. In the group session, the adolescent explored alternative ways to communicate with the father and identify elders whom she could approach for support.

### Giving and receiving support

Adolescents described how support from group members helped them to overcome their heart-mind problems. Support took various forms including, listening to and understanding each other's problems, creating a "cheerful environment" and suggesting potential solutions. One 16-year-old girl described how learning to become close to her group members helped her make other friends:

> *I studied in another school until Class 10 then came to this school. I didn't have any friends in the beginning. No one spoke to me […]. Later, I became very close to those in the group and I came to know everything about them. I came to know how to make friends and talk to people, and I applied the technique to other friends as well […]. It was very effective for me. Female 16.*

### Managing emotions and negotiating

When asked how IPT had helped them, many adolescents described the techniques they had learned to manage their emotions, specifically anger, loneliness and "tension". These techniques included "deep breathing", "counting backwards", playing with friends, drawing pictures and listening to music.

By managing their emotions, adolescents were able to better negotiate. A 15-year-old boy described how he had learned to wait until his friend was in a good mood before discussing a difficult issue with him ("If we explain while they are angry it might not work").

Adolescents also described how they implemented the communication skills they learned in the groups to help them negotiate. One adolescent who was upset about a dispute between her mother and aunt was able to negotiate their reconciliation. A 16-year-old girl described the "give to get" IPT communication strategy:

*If we have to take anything from people, we need to give them [something] as well. I learned those things. If I need money, I help my mother with her work then she happily provides me money. Female 16.*

### Context: Interpersonal and school factors affecting mechanisms and outcomes of IPT

Adolescents described how gender influenced their experience of IPT.

*When a guy is ruined and then recovers, he can get support from everyone. They accept him so nicely, saying that he got better even after being ruined and it's his life which he owns and has freedom of choice. However, once a girl is ruined, she never gets a chance to get better. That's why I think the session would benefit girls the most. Female, 16.*

Four adolescents suggested girls would benefit more from IPT than boys because they perceived that girls have more problems. A facilitator described the pressure on girls to uphold their family's reputation and how some would attempt suicide if they brought shame on their family. Girls explained that they have more household responsibilities than boys and are expected to earn for their families. They advised that their family and the community would think badly of them if they were late home so IPT sessions should be in the morning.

Age also influenced the way adolescents engaged in IPT. A trainer described how younger girls were shy which made it harder for them to share their problems and participate in discussions. Four adolescents thought IPT was more relevant for older adolescents because they are more likely to experience heart-mind problems due to being "under a lot of pressure" to study and fulfil their household responsibilities. However, adolescents believed IPT would not be helpful for older adolescents and adults who already have support and know how to solve their problems.

Two facilitators felt that the adolescents' family background (if their home was "deprived" or violent) was more important than age in determining the severity of their problems and potential benefits of IPT. One girl from the marginalised Dalit community described being discriminated against, isolated at school and having no one with whom to share her heart-mind problem prior to IPT. Adolescents found it difficult to overcome their problems where they had parents who abused alcohol and were verbally and physically violent, or where parents were stressed due to financial difficulty.

School-level factors affected the mechanisms and implementation of IPT. Trainers and supervisors conceptualised a role for schools in implementing IPT, providing a "supportive" and "encouraging environment" and mitigating stigma by normalising mental health problems. In schools where facilitators were able to mobilise support from staff, they could help even those adolescents experiencing extreme adversity.

*One case they found was abused by her brother-in-law. At that moment it was so tough for us to manage the case and then finally our facilitator found a very helpful teacher and through that network we were able to protect [the adolescent]. Supervisor.*

In contrast, in some schools facilitators found it difficult to work because staff did not trust them. In two schools, facilitators had difficulty arranging and accessing a room for the sessions. In one, facilitators were warned they were under CCTV surveillance and teachers did not like them asking adolescents' questions about their mental health. Adolescents did not want to attend IPT sessions in schools where staff and students "don't treat you well, tease and mock".

### Time: How the timing of IPT affected outcomes

The timing of the intervention, specifically the lack of pre-existing community mental health services in this setting, meant that IPT was met with both intrigue and suspicion by community members. Two adolescents said they "felt curious" about the programme and "wanted to see how it turned out". Human trafficking gangs were operating in the locality at the time of the study and facilitators said people were therefore wary of "outsiders". One mother described how she was worried IPT facilitators "might take [her son] somewhere and hurt him" and a 16-year-old girl thought facilitators might try to sell her.

Adolescents and facilitators described how mental health problems had always been stigmatised. A facilitator referred to community people as not "learned", meaning not educated or aware about mental health. Some people were unsupportive towards IPT and discriminated against those participating in it. Facilitators and supervisors described how a teacher referred to IPT groups as "psycho groups" and the adolescents as "psychos". Some adolescents described how their friends initially teased them about participating in IPT but later understood it was helpful.

Because of the lack of mental health services in the district, the programme team offered counselling to several parents of adolescents participating in IPT with severe mental health problems, and financial support towards a hospital admission for one parent. This service provision along with facilitators' efforts to build rapport with parents and school staff improved community attitudes towards the programme.

### Outcomes: Changes resulting from IPT

Adolescents' accounts implied that their social support improved through IPT, and this had positive impacts on their lives. Seven thought their studies had improved and one said this was because other group members helped with their homework. A 16-year-old girl described how she enlisted support from her brother to help stop her father from abusing alcohol and behaving aggressively towards her. Four adolescents mentioned improved friendships and four described improved relationships with family members. A trainer suggested this was because adolescents had learned how to reduce conflict.

Adolescents who learned how to manage their emotions also benefited from IPT:

*One of the [adolescents] was in depression […]. She did not talk to anyone in the beginning and used to get furious while talking […] so no one liked being close to her. Later, she began to gradually reduce her anger. She began to notice herself whenever she got angry. She calmed herself down whenever she realised her anger. […] Now she has a good relationship with everyone […]. Facilitator.*

Negative effects of IPT among adolescents related to feelings of sadness about the termination of the sessions and how they would miss seeing the facilitators and other group members. Adolescents had come to value the opportunity to share their problems and were concerned they would not have this in the future.

At the community level, implementing IPT in schools potentially exacerbated mental health inequities between in- and out-of-school adolescents:

*There was one [girl] in the village, she had left school. I wanted to bring her to the session and so I was planning to tell [the facilitator] but unfortunately it was too late and she committed suicide […]. I feel very bad about that. If only I'd taken her to the session I would have saved her life. […] I think you shouldn't just [recruit] the cases in the school but ask in the village as well. Female aged 16.*

### Rivalry: Competing commitments, responsibilities and opportunities affecting engagement in IPT

Adolescents had various responsibilities. Six adolescents and four facilitators described how adolescents had household work after school and on Saturday mornings, which impacted their ability to attend IPT sessions. One girl was engaged in paid work which affected her attendance at school.

Adolescents shared concerns about sessions overlapping with lessons and preparation for exams, which negatively affected their studies. One boy did not attend sessions because of an inter-school competition. One boy thought it would be better if sessions could be before school so adolescents would not have to leave their school friends.

Ill health was another reason for missing sessions: either the adolescents were sick themselves or they had to care for sick family members. Family events and festivals also impacted attendance. A 15-year-old boy described how his father was hospitalised in another district. He had to move away with his family to be closer to his father.

Many adolescents lived in remote villages several hours' walk from school. One facilitator described how it was difficult to run sessions because adolescents were tired from travelling to and from school. During the programme there were landslides and floods which reduced adolescents' ability to travel. Unreliable phone reception and internet access affected communication between sessions which compromised engagement and rapport between adolescents, their parents and the facilitators.

### Emergence: Novel or unexpected adaptations to IPT

Facilitators and adolescents described the impact of the COVID pandemic on the sessions and the necessary adaptations which were made. Lockdown restrictions meant group meetings had to stop so facilitators conducted termination sessions on an individual basis by phone. Facilitators and adolescents lamented the opportunity to meet in person. One facilitator described how he missed talking to adolescents and seeing the other facilitators who had become his friends.

During the programme, many adolescents disclosed suicidal thoughts and behaviours. Facilitators described how they managed these cases in line with the study protocol but that the interactions were stressful, made them anxious and sometimes they felt "triggered". One facilitator described how he "couldn't focus, felt heavy and became tired". He was scared that the adolescent might attempt suicide despite his efforts. Supervisors mobilised to support facilitators practically and emotionally through supervision, training on a standardised protocol and referral of high-risk cases to a trained counsellor. Facilitators also supported each other: one facilitator described a "sense of teamwork". For example, when one facilitator had to take leave to care for a family member the other facilitators "helped a lot" by taking over his responsibilities.

## Discussion

Our findings shed light on the complexity of a psychological intervention for adolescents in rural Nepal. We found evidence for potential mechanisms including sharing and solving interpersonal problems, giving and receiving support, managing emotions and negotiating. Adolescents' experience of IPT was influenced by their age and gender, the nature of their problems, barriers to participation and family support. Schools had a pivotal role in the success of IPT in terms of providing resources and a supportive environment for adolescents and facilitators.

Several CMOCs emerged from the analysis. First, girls who do not have support from their family and are unable to solve their heart-mind problems (context) will receive support, advice and techniques from other IPT participants and facilitators that they can use to solve their problems (mechanisms) which will lead to reductions in depression (outcome). Wider literature supports the importance of gender as a contextual factor. In a scoping review of adolescent depression treatment moderators (baseline variables associated with differential outcomes between treatment groups) out of 13 trials four reported significant effects of female gender in multivariable analysis (Courtney et al., 2022). In our feasibility study of IPT in Nepal (albeit a small uncontrolled study, n = 62) we observed improvements in both genders, but we observed a greater improvement in depression among boys. We hypothesised that this might be because girls had less support from their families and agency to make positive changes in their lives compared to boys. Research is needed to understand how IPT facilitators navigate agency in gendered systems of power, and whether girls who challenge these systems experience negative consequences (Budge and Moradi, 2018).

A second CMOC concerns age and developmental stage: older adolescents (context) will feel confident to share their heart-mind problems and participate in discussions where they give and receive support from IPT participants and facilitators (mechanism) which will reduce their depression (outcome). In our feasibility study in Nepal, both younger (13–14 years) and older adolescents' (15–19 years) depression scores improved after IPT, but it is possible different mechanisms were at play. A trial of IPT among adolescents in the US found larger effects among older adolescents (Mufson et al., 2004). Theorising, older adolescents with more developed executive function and social cognition may have a greater ability to comprehend and implement IPT techniques and strategies compared to younger peers (Blakemore and Choudhury, 2006; Courtney et al., 2022). Younger adolescents who are less emotionally developed may find it difficult to recognise, understand, communicate and regulate their emotions, and may benefit from more parent involvement in treatment (Fonagy et al., 2015). However, the developmental psychology and neuroscience research

on which these theories are based has mainly been done in a handful of high-income countries. Research is therefore needed to understand how generalisable findings are to the rest of the world, including Nepal.

Adversity is the focus of the third CMOC. Adolescents who are experiencing domestic violence and/or have a parent who abuses alcohol (context) will have limited ability to implement problem-solving techniques that substantially improve their home environment (mechanism), which will diminish any improvement in depression (outcome). Adolescents experiencing adversity may be practically unable to change their home environment, especially in rural Nepal where there are no mental health services or support for victims of violence. Moreover, research suggests adversity causing deprivation and threat has implications for adolescent development, including emotional learning, executive function and problem-solving (Jagasia et al., 2024). This may affect how adolescents engage with and use IPT.

The last CMOC relates to school climate, defined as the quality and character of a school. Adolescents attending schools where staff care about student wellbeing and do not stigmatise mental illness (context) will receive emotional and practical support from staff that enables them to solve their heart-mind problems (mechanism) leading to reductions in depression (outcome). Our findings also suggest that schools could help to optimise results by allocating a quiet, private space for group sessions that promotes sharing and confidentiality. There is wider evidence for the effect of school climate on depressive outcomes. For example, in a trial of a whole school health programme in Bihar, India, a climate characterised by supportive and engaged staff-student relationships, a sense of belonging and student participation in school events predicted lower rates of depression (Singla et al., 2021). Our study highlights the importance of early consultation with and formative research in schools to assess readiness for mental health intervention and the need for psychoeducation and training for staff and students, with implications for future school-based mental health policy.

### Strengths and limitations

Strengths of the study include the purposive diversity of the sample, and triangulation of evidence from interviews with facilitators, supervisors and master trainers. Interviews with adolescents and facilitators happened immediately after the intervention, but interviews with supervisors and trainers happened two years after the intervention and this is a limitation. Interviews with adolescents and facilitators did not directly ask about each of the components of the VICTORE checklist and therefore data related to some components are richer than others. We had difficulty categorising data related to intervention mechanisms and adaptations using the checklist. Ultimately, we captured mechanisms under implementation and adaptations under emergence which may not be what was intended by Pawson but was logical for this study. An alternative qualitative analysis approach is dimensional analysis, a variant of grounded theory, which has been used to explore context, conditions, process and consequences in other school-based complex interventions (Bonell et al., 2022). Realist research on psychological interventions is an emerging field so there is limited information to recommend one analytical approach over another. For several reasons, we opted to use framework analysis informed by the VICTORE checklist over dimensional analysis: (i) data from facilitators and adolescents were not specifically collected to elucidate theory on context, mechanisms and outcomes, and the components of the VICTORE checklist provided a broader framework

within which to analyse the data compared to dimensional analysis; the checklist enabled us to directly apply realist principles to our analysis rather than having to translate findings through a grounded theory lens; (iii) in our experience, it is easier to involve multiple researchers in a framework analysis and more accessible to less experienced qualitative researchers compared to grounded theory (Gale et al., 2013).

## Conclusions

Our study demonstrates the relevance of realist research for psychological intervention. We formulated hypotheses about mechanisms, as well as contextual interpersonal and school factors that may interact with these mechanisms to bring about reductions in depression. Research is needed to refine and test our CMOCs in Nepal and explore their relevance in other settings.

**Open peer review.** To view the open peer review materials for this article, please visit http://doi.org/10.1017/gmh.2024.127.

**Data availability statement.** Due to the sensitive nature of the topic of this study and the difficulties of fully anonymising the transcripts data are not suitable for sharing.

**Author contribution statement.** KA, KRC and CB conceptualised and designed the study, KA and KRC analysed the data, KA and KRC wrote the first draft of the paper, and all authors commented on the draft and approved the final version.

**Financial support.** The work was supported by the UK Medical Research Council, the National Institute for Health Research, and the Department for International Development (KRC, grant reference MR/R020434/1) and a UK Research and Innovation Future Leaders Fellowship (KRC, grant reference MR/W00285X/1).

**Competing interest statement.** Conflicts of Interest: None.

**Ethics statement.** Nepal Health Research Council (637/2018) and King's College London Research Ethics Committee (KCL REC, RESCM-18/19–8,427) approved the feasibility study (including qualitative interviews with adolescents and facilitators). We obtained further ethical clearance for interviews with IPT supervisors and master trainers in 2022 from KCL REC (MRA-22/23–34,693). We obtained informed consent for all participants and parental consent for adolescents under 18.

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
