## [Reviewer Report]

The findings are insightful. Further questions could explore how these components impact the effectiveness of IPT when incorporated. It would be interesting to examine how effective IPT was in reducing depression among adolescents in the main study (Rose-Clarke et al., 2020) and identify any synergy between the two. Do the findings in the main study align with or contradict some of the results presented in this article?

---

## [Reviewer Report]

Following a realist approach with a clear framework, the present manuscript provides a well nuanced illustration of the complexities involved in the real life implementation of an intervention (i.e. group IPT) and understanding how it works. The qualitative data are presented in a coherent and systematic way, providing the reader with a multilayered perspective on the experiences of study participants. The findings of the study are adequately contextualized and seem pertinent to our understanding of how the intervention works, who it benefits the most and the least, and the conditions/circumstances that optimize or hinder it. (Contextual factors, such as school culture and staff attitude toward mental health, are but one example of how an intervention cannot be separated from the context in which it’s deployed.) Future research can build on the present findings to formulate and test hypotheses in a more informed way.

---

## [Reviewer Report]

Thank you for the opportunity to review this interesting article, and to the authors for their work. Overall, this article has several strengths, and I enjoyed reading this paper. These strengths include most notably: 1) the practical and promising analytic methods used, and 2) the interesting implications of this study.

I have only a few minor comments on this article, alongside a note to please re-check spelling, grammar, and punctuation:

Abstract:

Because most readers won’t be familiar with the methods used in this paper (they are quite novel), the authors should add briefly the fact that analyses were qualitative (e.g., the qualitative analytical framework) so that there is no doubt when reading the results that they are based on qualitative rather than quantitative analyses.

The findings are interesting, clear, and make sense.

Introduction:

It’s unclear why CBT is mentioned in the Introduction when it is not otherwise so relevant to the article. The mentions of CBT could probably be deleted.

The last two paragraphs of the introduction would be made stronger if they included more concrete descriptions of the methods, still without giving unnecessary details. For example, it again isn’t fully clear that the authors seem to be applying a framework-based codebook to assess themes present in intervention feedback interviews.

It would be helpful to include in the introduction a brief discussion of past research on IPT effect moderation, including discussion of any established moderators.

Methods:

In the sub-section “Setting” the authors mention when data on depressive symptoms were collected, which seems irrelevant to the present manuscript, and is a bit confusing as it left me expecting more information on the measure used and results of this data collection.

How were the 16 individuals selected from the 62 total study participants to take part in the qualitative interviews?

Table 2 is very helpful! However, in the analysis section, it is not entirely clear how these codes were applied to the transcripts to generate the information in the results section. Did the two authors discuss discrepancies? Were transcripts double or single-coded? Is there anything else important to note about the coding process?

Results:

Generally, the results are interesting and well-written.

One issue is that sometimes, numbers are provided for how often themes were discussed (e.g., four adolescents, one participant) but other times these numbers are not mentioned. Numbers of participants should consistently be mentioned so that readers know how common each theme was across transcripts.

Discussion:

The discussion of findings related to gender is very interesting. This will certainly be an important area for future research!

The authors mention a two-year gap between the study and interviews with trainers and supervisors; was there a similar gap for the interviews with adolescents? If not, that is worth clarifying here.

Dimensional analysis is mentioned as an alternative in limitations, but it is unclear whether this method would have been preferable, or why not using dimensional analysis might be a limitation.

---

## [Reviewer Report]

I have no serious remaining concerns about this manuscript. It is still interesting and valuable, and is now clearer than in the original version as well. Congratulations to the authors on this valuable work!